# Never Ending Reasoning and Learning: Opportunities and Challenges

**Sriraam Natarajan[1], Kristian Kersting [2]**

[1] University of Texas at Dallas, USA
[2] TU Darmstadt, Germany
Sriraam.Natarajan@utdallas.edu, kersting@cs.tu-darmstadt.de

## Abstract

Inspired by the motivation behind the Never-Ending Language Learner (NELL), a continual learning system that reads the web, we propose the never-ending reasoning and learning paradigm and one instance: the Never-Ending Reasoner and Learner (NERL), which continuously learns and reasons with causal models by actively interacting with domain experts. NERL necessitates tight synergistic interaction between different communities—continual learning, causal modeling, statistical relational AI, and human-allied AI communities. We motivate NERL using the real, high-impact problem of global mitigation of adverse pregnancy outcomes, present the challenges in this system, and highlight the potential opportunities that provide for interdisciplinary collaborations.

## Never-Ending Reasoning and Learning

Building life-long learning systems that can continuously learn from, interact with, collaborate with, and potentially teach domain experts has been a long, cherished goal of AI research. To this effect, inspired by the never-ending language learner (NELL) (Mitchell et al. 2015) out of Tom Mitchell's group, we present the *never-ending reasoning and learning* paradigm and one instance: the Never-Ending Reasoner and Learner (NERL). NERL serves to combine different yet related fields of AI namely, causality (Pearl 2009a), continual learning (van de Ven and Tolias 2019), statistical-relational AI (De Raedt et al. 2016) and human-allied learning. In this short note, we motivate the need for combining these fields and provide a list of challenging problems and opportunities of this combination.

To challenge current AI and to motivate NERL, consider the task of developing strategies to mitigate adverse pregnancy outcomes. Building an AI system that is generalizable across the world's population requires multiple research areas including the ones mentioned above. Specifically,

- most medical knowledge is causal,

- the clinical knowledge and data (due to devices such as wearables, sequencing improvements and increase in multimodal data) continuously evolve,

- there is continuous interaction with domain experts (clinicians and providers),

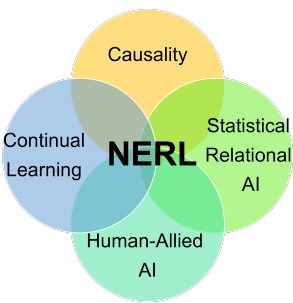

Figure 1: Never-ending reasoning and learning

- a rich relational structure is present both in data and knowledge (for example, family history), and

- a need to generalize/transfer across different levels of abstraction (at the world population level, specific to a certain race, or even to a particular group, say south Indians).

The above example of mitigating adverse pregnancy outcomes clearly illustrates the need to develop synergistic interactions between different sub-fields of AI and machine learning. We specifically focus on an integrated research platform that combines four different sub-fields, cf. Fig. 1. First is causality that provides theories and tools for modeling, learning and reasoning about causal relationships that exist in the domain (clinical knowledge in our example, for instance, higher BMI can cause gestational diabetes). The second is continual learning that allows for learning from a continuous, multimodal stream of data (for instance, learning from clinical studies, then from electronic health records, and augmenting the model from wearable data such as sleep, heart rate, activity levels etc.). The third is statistical relational AI that allows reasoning over different levels of abstraction (for instance, population/sub-population levels) by using a higher-order representation such as first-order logic while allowing for modeling uncertainty using statistical methods including but not limited to probability theory. The final sub-field is human-allied AI that allows for learning and reasoning with human experts in the loop (in our example, healthcare providers).

To meet the challenge outlined, we start from NELL. NELL is a continuous learning system that aims to "read the web" by crawling millions of web-pages and continu-

ously builds concepts and facts. It represents these facts using probabilities to reflects its beliefs based on the evidence it collects. It is similar to the mega reasoning engine Cyc by Cycorp (Lenat 1995) that has accumulated knowledge for several decades. We are inspired by these two systems in proposing the NERL system that allows for learning from multi-modal data, evolving knowledge and finally, regular interaction with the human-in-the-loop. In the next section, we briefly outline the challenges for this system and hope to encourage participation from the broader community on solving these challenges.

## NERL: Challenges & Potential Opportunities

When developing a NELL and Cyc equivalent in a clinical/healthcare task, there are several non-trivial challenges that require the collective intellect/effort of the AI community to bridge the different research areas. We present a (not exhaustive) list of these challenges now.

**(1) Continuous learning from multimodal data.** Several real domains such as healthcare require models that can be learned from different data sources – images, text, EHR, genetic data and potentially wearable data (raw sensor) data. As we will discuss later, since different knowledge sources exist, simply combining all these data types to a single compressed representation leads not only to a loss of information but also does not allow for providing rich, faithful knowledge that also continuously evolves. There is a need for developing learning methods that could allow for a seamless integration (potentially in a hierarchical manner) of the multimodal data.

**(2) Evolving causal knowledge.** One key challenge for NERL is not just that data changes over time but the causal knowledge itself changes. This could be due to new discoveries made, new treatments developed and most importantly, some newer hidden confounders identified. The other important aspect here is that this knowledge can be quite noisy or uncertain. Relearning the entire model is impractical and, hence, a continuous updating of the causal knowledge is necessary in real-time deployment.

**(3) Using higher-level knowledge to learn lower-level systems.** Given the surge of deep learning methods in vision and NLP (and their combination such as Dall-E2 (OpenAI et al. 2018)), it is abundantly clear that the knowledge for these systems are more natural and faithful at a fairly higher-level, i.e., at the level of objects and relations between them rather than at specific individual instances. This abstract knowledge should then be employed appropriately during learning and reasoning. Leveraging this continuous, evolving knowledge is challenging to NERL.

**(4) Causal reasoning at different levels of abstraction.** Most importantly, the knowledge tend to be mainly causal specifically in healthcare domains. Such causal knowledge can also be provided in different levels of abstractions and can be represented using causal independence or context-specific independence statements (Pearl 2009b). For instance, one could specify that east Asian women have a higher incidence of gestational diabetes and older women are at a higher-risk of pre-term birth. This knowledge can then be superceded by computing a polygenic risk score of a particular condition for the given woman. The NERL system must be able to reason with such rich knowledge and adapt its reasoning according to the new evidence and knowledge.

**(5) Effective human-machine communication and interaction.** An important aspect of NERL is that it continuously interacts with the human to learn from, collaborate with and potentially teach the expert. Hence, it is necessary that the vocabulary has to evolve as well. For instance, when a new discovery is made or a new condition is diagnosed, the machine and the human should use the same set of concepts to describe this. Continuous evolution of concept dictionary is crucial in achieving this goal. There is a need for an abstract meaning representation that allows for continous learning and refinement of learned concepts.

## Discussion

In this short position paper, we motivated the need and potential opportunities of NERL, a never ending reasoning and learning system. The motivations were built based around a concrete use case in healthcare, that of improving pregnancy outcomes across the world. Our key observation is the need for synergistic interactions between different research communities inside AI. We identified a non-exhaustive list of four such communities - continual learning, causal learning and reasoning, statistical relational AI and human-allied AI communities. Integration with NLP, vision and other communities is potentially interesting and allows for developing a truly interdisciplinary research agenda inside AI.

## Acknowledgements

Sriraam gratefully acknowledge AFOSR Minerva award FA9550-19-1-0391. Kristian acknowledges the support of the Hessian Ministry of Higher Education, Research, Science and the Arts (HMWK) in Germany, project "The Third Wave of AI".

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
