# OpenReview forum: "Never Ending Reasoning and Learning: Opportunities and Challenges"
_AAAI.org/2023/Bridge/CCBridge — AAAI23 Bridge Continual Causality_

### Official Review · Reviewer_26ED · 2022-11-24
**Interesting read, excited to see the results of this project.**

**Rating:** 9
**Confidence:** 3

**Review:**

I found this an interesting position paper that got me excited to follow the progress on this project. The paper nicely motivates the Never-Ending Reasoning and Learning paradigm both at a conceptual level, and with the adverse pregnancy outcome example also at a somewhat more practical level.

I think the paper connects well with the bridge topics.

---

### Official Review · Reviewer_L9jc · 2022-11-30
**Suggestions for Improvements**

**Rating:** 8
**Confidence:** 5

**Review:**

This is a well written position piece. I think motivating a bit more strongly the continual learning application would be helpful in terms of why doing this without continual learning (just growing a database and periodically retraining) would be impractical. I do think that the specific example given would be interesting from a research perspective; however, for medical applications continual learning is pretty tricky since medical regulatory agencies really like to have very characterized behavior of systems with locked down systems assessed in well designed studies. I think the multi-modal nature of the data mentioned later could motivate continual learning, but it wasn't really clear that the data would be high-dimensional in the introductory example compared to what's described later.

Another angle specific to medicine is patient health information. Continual learning could be used to motivate being able to update the system without retaining a veridical copy of the training data, which may not be allowed due to data access rights or patient privacy reasons.


Typo in discussion: thee --> the

---

### Decision · Program_Chairs · 2022-12-05

**Decision:**

Accept

**Comment:**

Accept - Oral

This position paper proposes the never-ending reasoning and learning paradigm, relevant to many disciplines including the bridge topics: causal modeling and continual learning. The topic is thus well aligned for the bridge program. We suggest that the authors use the additional space in the camera-ready version to integrate the reviewers’ comments and concerns, including additional motivation of the continual learning problem in this setting.